# A Novel Genotype and First Record of *Trypanosoma lainsoni* in Argentina

**DOI:** 10.3390/pathogens9090731

**Published:** 2020-09-04

**Authors:** Anahí G. Díaz, Paula G. Ragone, Fanny Rusman, Noelia Floridia-Yapur, Rubén M. Barquez, M. Mónica Díaz, Nicolás Tomasini, Patricio Diosque

**Affiliations:** 1Unidad de Epidemiología Molecular (UEM), Instituto de Patología Experimental, Universidad Nacional de Salta-CONICET, 4400 Salta, Salta, Argentina; anahi1990gd@gmail.com (A.G.D.); p_ragone@yahoo.com.ar (P.G.R.); fannyrusman@gmail.com (F.R.); narfy89@gmail.com (N.F.-Y.); 2Programa de Investigación de Biodiversidad Argentina (PIDBA), Facultad de Ciencias Naturales e Instituto Miguel Lillo, Universidad Nacional de Tucumán-CONICET, 4000 San Miguel de Tucumán, Tucumán, Argentina; rubenbarquez@csnat.unt.edu.ar (R.M.B.); mmdiaz@lillo.org.ar (M.M.D.)

**Keywords:** *Trypanosoma lainsoni*, *Leopardus geoffroyi*, *Calomys* spp., 18S rDNA genes, gGAPDH genes, morphology, transmission electron microscopy

## Abstract

Trypanosomes are a group of parasitic flagellates with medical and veterinary importance. Despite many species having been described in this genus, little is known about many of them. Here, we report a genetic and morphological characterization of trypanosomatids isolated from wild mammals from the Argentine Chaco region. Parasites were morphologically and ultrastructurally characterized by light microscopy and transmission electron microscopy. Additionally, 18s rRNA and gGAPDH genes were sequenced and analyzed using maximum likelihood and Bayesian inference. Morphological characterization showed clear characteristics associated with the *Trypanosoma* genus. The genetic characterization demonstrates that the studied isolates have identical sequences and a pairwise identity of 99% with *Trypanosoma lainsoni*, which belongs to the clade of lizards and snakes/rodents and marsupials. To date, this species had only been found in the Amazon region. Our finding represents the second report of *T. lainsoni* and the first record for the Chaco region. Furthermore, we ultrastructurally described for the first time the species. Finally, the host range of *T. lainsoni* was expanded (*Leopardus geoffroyi,* Carenivora, Felidae; and *Calomys* sp., Rodentia, Cricetidae), showing a wide host range for this species.

## 1. Introduction

Trypanosomatids are unicellular parasites characterized by a single flagellum. This family belongs to the class Kinetoplastea, named for the presence of a large mitochondrion with numerous copies of circular DNA, organized in a compact structure called kinetoplast [1]. This class is composed of 12 parasitic genera with numerous species that can each parasitize either vertebrates, invertebrates, or even some plants [2,3,4]. Among them, *Trypanosoma* is a monophyletic group that infects all classes of vertebrate hosts [5,6,7], shows different morphological types during their dixenous life cycles, and are transmitted by blood-sucking invertebrates [8,9].

Different phylogenetic groups within this genus have been delimitated by phylogenetic analyzes of the 18S rDNA and glycosomal glyceraldehyde phosphate dehydrogenase (gGAPDH) genes [5]. The most studied species within the genus *Trypanosoma* are those of medical and veterinary importance, such as *Trypanosoma cruzi, T. brucei, T. vivax*, and *T. evansi*. Conversely, very little is known about the other species of the genus *Trypanosoma*. An example is the phylogenetic group of trypanosomes of lizards and snakes (e.g., *T. serpentis*, *T. cascavelli*, *T. varani*) [7]. After first descriptions and based on phylogenetic analyses, this group was expanded to also include parasites from rodents and marsupials, i.e., *T. freitasi* and *T. lainsoni.* Consequently, the clade was renamed to lizard and snake/rodent and marsupial (LSRM) [7,10,11,12,13]. Particularly, *T. lainsoni* was represented by a single isolate from a caviomorph rodent of the genus *Mesomys* in the northern state of Amazona, Brazil.

South America has a great number of trypanosome species, parasitizing a wide diversity of hosts. However, most of the reports on isolations are registered in Brazil [7,10,11,12,13,14,15], while in Argentina, just three species were reported: *T. cruzi* (the agent of Chagas disease), *T. evansi*, and *T. vivax* [16,17,18,19].

In this study, we analyze the phylogenetic relationships of trypanosomatids isolated from wild mammals from the Copo National Park, in the Argentine Chaco region. Particularly, we described a new genotype of *T. lainsoni* obtained from a *Leopardus geoffroyi* (Carnivora, Felidae) and from two rodents by analyzing the rDNA and gGAPDH genes. In addition, we provide the first cell ultrastructure description for this species.

## 2. Results

### 2.1. Light and Transmission Electron Microscopy Classified the Three Isolates to the Genus Trypanosoma (Trypanosomatidae)

Light microscopic observations of in vitro cultures showed that the three isolates were morphologically compatible with trypanosomatids. These isolates presented cells with an average length of 33.4 µm and tapered shapes, with a single flagellum, a nucleus, and a kinetoplast (Figure 1A). The forms identified in culture were: trypomastigotes (central nucleus with a kinetoplast and posterior flagellum) (Figure 1B), epimastigotes (central nucleus with the kinetoplast and the flagellum anterior to nucleus) (Figure 1C), and spheromastigotes (nucleus, central kinetoplast, and the flagellum surrounding the body) (Figure 1D).

In the culture conditions, a great variation was observed in the length of epimastigotes throughout the exponential phase, from 25.6 to 72.5 µm (Figure 2A). In addition, we observed replicating epimastigotes, identified by the presence of two nuclei, two flagella, and a single kinetoplast (Figure 2B). On the other hand, the trypomastigotes were observed at the end of the exponential stage and were smaller than epimastigotes (Table 1; Figure 2C). Finally, transitional forms between epimastigotes and trypomastigotes were also detected (Figure 2D). The size of these last parasites was variable (Table 1).

In concordance with light microscopy images, the samples analyzed by TEM (Transmission Electron Microscopy) revealed typical structures of *Trypanosoma*, such as a kinetoplast, flagellar structures (flagellar pocket, basal bodies), many vesicular bodies, acidocalcisomes, reservosomes, lipid bodies, and a cytostome. Most of the observed forms were epimastigotes (Figure 3A,B). In these forms, a large nucleus can be recognized in a central position with an average length of 1.29 ± 0.17 µm and an average width of 1.03 ± 0.13 µm (Figure 3C). In addition, a rod-shaped kinetoplast was observed without a clear differentiation of ridges (Figure 3D) and an average length of 0.94 ± 0.32 µm. The kinetoplast was anterior and next to the nucleus and a single asymmetrical flagellar pocket. This structure had a probasal body on the bulky side and a basal body on the other one (Figure 3E). Furthermore, the cytostome (Figure 3F) was located near the flagellum and the kinetoplast. A great number of vesicles of variable size and dark coloration, acidocalcisomes, and rounded lipid bodies were also observed (Figure 3G). Finally, reservosomes (rounded structures) were visualized along some epimastigotes (Figure 4A,C).

### 2.2. The Isolated Trypanosomes Belong to Trypanosoma lainsoni and Constitutes a Novel Genotype Different to the Brazilian Isolate

The alignment of the first sequences obtained for the 18S rDNA gene (610 bp) revealed that the three isolates Ca37, Ca47, and Le29 are completely identical. This was subsequently corroborated with the sequence of the gGAPDH gene (528 bp). These sequences were analyzed by BLAST and they were similar to *T. lainsoni*, although not identical (Appendix A).

The complete 18S rDNA gene sequence was obtained from Le29 (2659 bp) and used in the following analyses. Like the first sequenced fragment, the Le29 isolate was found to have 99% shared identity with the *T. lainsoni* species for the 18S rDNA complete gene (Appendix A). Furthermore, the gGAPDH and 18S rDNA genes of the Le29 isolate had pairwise identity greater than 90% with the sequences published for the *T. freistasi*, *T. gennarii*, and *T. casacavelli* species [7,10].

The high pairwise sequence identity between Le29 and the only *T. lainsoni* isolate reported in Brazil indicated that they belong to the same species. However, we made a phylogenetic analysis in order to address such genetic distance in the context of other distances in the LSRM phylogenetic group. The maximum likelihood and Bayesian methods generated phylogenetic trees with topologies similar to those reported for the genus *Trypanosoma* (Figure 5, Appendix A). The trees for the gGAPDH gene and for the concatenated alignment showed very short genetic distances (0.6%) between the sequences of the Le29 isolate and the Brazilian *T. lainsoni* (Figure 5 and Appendix A), which prevent us separating them into different species (Appendix A).

## 3. Discussion

In this study, the morphological and genetic characterization of three trypanosomatid isolates was performed. The parasites were isolated from mammalian hosts in the Copo National Park, Santiago del Estero province, in the Argentine Chaco region. They were identified as *Trypanosoma lainsoni*, a species previously described from the blood of a rodent (genus *Mesomys*) in Brazil [13] and later included within the LSRM clade [10]. This report represents the first isolate of this species recorded for Argentina. Furthermore, this paper constitutes the second report of *T. lainsoni* and the first cell ultrastructure description for the species. Additionally, we found *T. lainsoni* in the Order Carnivora (*Leopardus geoffroyi*)*,* showing a wider host range for the species.

The morphology observed under light and electron microscopy showed structures according to the Kinetoplastea class and the Trypanosomatidae family. In addition, the presence of stages such as epimastigotes and trypomastigotes, as well as a great variety of intermediate forms, were compatible with those described for the genus *Trypanosoma*. In the same way, typical internal cell structures of *Trypanosoma*, as structures related to the feeding of cells such as the cytostome and the reservosomes. These organelles are characteristic of trypanosomes, where the macromolecules used during the metacyclic process from epimastigote to trypomastigote are stored [20]. On the other hand, the kinetoplast in the epimastigote stage presented a rod-shape, similar to kinetoplasts of trypanosomes in the same phylogenetic group (LSRM clade) [7,21] and a similar thickness compared to other trypanosomes [7].

The morphometric data revealed that *T. lainsoni* isolates described here have similar sizes than those from the Amazon region in Brazil (36.2 µm and 33.4 µm, respectively). Based on this size, the Amazonian isolate was considered as part of the subgenus *Megatrypanum* [10,12,22]. Consequently, *T. lainsoni* was granted the status of a new species because there was no record of *Megatrypanum* isolates from a rodent caviomorph [13]. Afterwards, other studies reaffirmed the status of new species according to molecular data [10]. In addition, the presence of *T. lainsoni* in mammalian hosts is reinforced. However, it is important to note that the criterion to define species based on the host is weak because it is known that many trypanosomes do not usually have host specificity. Consequently, phylogenetic techniques are more useful to define species [11], at least in trypanosomatids, and prevent us from defining a new species based only on the host where the parasite was isolated. The genetic characterization determined that the three isolates from the Chaco region share 100% similarity in the partial sequence of the 18S rDNA gene and the complete sequence of the gGAPDH gene, indicating that they are the same species and the same genotype, although we cannot discard that they may differ in other regions of the 18s rDNA or in other genomic regions. However, the isolates that we describe here are a different genotype from those previously described from the Amazon region, based on the sequences studied. The latter is not surprising considering the geographical distances and the different hosts from which the isolates from Brazil and Argentina were obtained.

The phylogenetic tree built with the concatenated 18S rDNA and gGAPDH genes raised questions about the taxonomic criteria currently used to classify parasites isolated from different hosts as different species. The concatenated tree showed a low phylogenetic distance between some species. Such distances were less than the distance found between strains belonging to a same species, such is the case of *T. cruzi* strains, which ranges from 0% to 1% (Appendix A). An example is observed among the isolates of the species *T. serpentis* and *T. cascavelli*, which have a phylogenetic distance of 0.2%, less than that found between the Le29 isolate and the *T. lainsoni* isolate from Brazil (0.6%) (Appendix A). In this sense, we consider that at least in some cases, the criteria used to define new species within the genus *Trypanosoma* are unclear. Further analysis and consensus would be required in order to clearly define how much genetic distance should be considered sufficient to differentiate species in this group. Consequently, because host and distribution range appear not to be useful for defining *Trypanosoma* species in many cases and the genetic distance between Le29 and *T. lainsoni* did not exceed 1% (the distance between some *T. cruzi* strains), we consider that Le29 should be considered as a different genotype of *T. lainsoni.* No vector was identified in the park despite capture efforts with light traps. Suspicious vectors could be those already reported for the LSRM clade, such as flies, sand flies, and various species of ticks [21,23,24,25,26,27]. On the other hand, the oral route cannot be ruled out since it is probably the most frequent mechanism of transmission of trypanosomes in the wild cycle [28]. Taking into account that the diet of *L. geoffroyi* is mainly composed of rodents of the *Akodon* and *Calomys* genera [29], we think that oral transmission would be a possible route of circulation of *T. lainsoni* in the studied geographic area.

Finally, it would be interesting to elucidate if *T. lainsoni* naturally infects domestic animals or even humans. In the same way, taking into account that *T. cruzi* (the causal agent of Chagas disease) is highly endemic in the Chaco region where *T. lainsoni* was isolated, it would be useful to know whether *T. lainsoni* infection can interfere with the serological diagnosis of *T. cruzi* infection. The latter could be addressed through studies of experimental infections in mice.

## 4. Materials and Methods

### 4.1. Wild Animal Capture

The parasites were isolated from *Leopardus geoffroyi* (Carnivora, Felidae) and two *Calomys* sp. (Rodentia, Cricetidae). These hosts were captured in Copo National Park, Santiago del Estero, Argentina, within the Chaco region (25°58′ S, 61°53′ W). Small mammals were trapped using Sherman traps, while medium size mammals were captured using Tomahawk traps. All animals were anesthetized with isofluorane and blood samples were obtained by cardiac puncture in small mammals and by venipuncture in medium size mammals. The animals were manipulated in concordance with the recommendations of the Institutional Committee for the Care and Use of Experimental Animals of Argentina. Ethical approval number 31/2014, date 10/21/2014, National Parks Administration, Argentina.

### 4.2. Isolation and Culture of Trypanosomes

Trypanosomatids were isolated from blood samples. Samples of 500 µL were placed in biphasic medium composed of a solid phase (4% agar supplemented with rabbit blood) and a liquid phase composed of Liver Infusion-Triptose medium (LIT) supplemented with 20% fetal bovine serum, hemin 20 µg/mL, penicillin 100 IU, and streptomycin 100 µg/mL under shaking at 25 °C. After corroborating the presence of trypanosomatids, the isolates were maintained in the laboratory in the same medium. One isolate was obtained from *Leopardus geoffroyi* (named Le29) and two from *Calomys* sp. (named Ca37 and Ca47).

### 4.3. Morphological Characterization

#### Light and Electron Microscopy

Daily culture samples of each isolate were observed under light microscopy at 1000× to identify and measure different parasite forms. Additionally, a smear on a glass slide per sample was fixed with fetal bovine serum and subsequently stained using the May Grünwald and Giemsa staining method and observed at 1000×. Different parasite forms were compared with those described for cultivated parasites of the *Trypanosoma cruzi* species [30].

For transmission electron microscopy, a sample from the exponential growth phase of the Le29 isolate was taken and fixed with 2% glutaraldehyde and analyzed by the Service of the Electron Microscopy Center of the Faculty of Veterinary Sciences of the National University of La Plata, Argentina. The obtained images were processed and compared with those available in literature.

### 4.4. Molecular Characterization

#### 4.4.1. PCR and Sequencing of 18S rDNA and gGAPDH

Total DNA was extracted from each isolate with the commercial DNA Puriprep T-kit (INBIO HIGHWAY; Buenos Aires, Argentina). The 18S rDNA and gGAPDH genes [31,32] were amplified by PCR. The amplification of the 18S rDNA gene was carried out in two steps. First, a 610 bp fragment was amplified using the 18sLeft and 18sRight primers (Appendix A). Secondly, the entire gene of approximately 2000 bp was amplified using the 18sp3pL and 18sp3pR primers (Appendix A). Cycling conditions were: 95 °C for 2 min, followed by 35 cycles at 94 °C for 30 s, 57 °C for 30 s, 72 °C for 2 min, and 72 °C for 20 min. On the other hand, amplification of 528 bp of the gGAPDH gene was performed using the ggapdh2F (5′-TGCACGGAARRTTTAAGCAC-3′) and ggapdh2R (5′-GAGCTTCGGTTGTCGTTGAT-3′) primers. Cycling conditions were: 95 °C for 2 min, followed by 35 cycles at 94 °C for 20 s, 57 °C for 30 s, 72 °C for 1 min, and 72 °C for 10 min.

Each amplicon was purified and sequenced by means of an automatic ABI3130 capillary sequencer (Applied Biosystems; Waltham, MA, USA), provided by the Sequence Service Cerela-CONICET (Tucumán). Internal primers were used to optimize the sequencing of 18S (Appendix A).

#### 4.4.2. Data Analysis

The obtained electropherograms were manually edited using Chromas v2.6.5 (https://technelysium.com.au/wp/; Technelysium Pty Ltd., South Brisbane, Australia). The edited sequences were uploaded to the GenBank (codes in Table 2) to be compared with other available trypanosome sequences (Appendix A) by using the version 7.0.26 of MEGA (MEGA Software, PA, USA) [33]. For the alignment of the 18S rDNA gene, the Gblocks tool (http://phylogeny.fr) was used to eliminate the hypervariable regions.

The following alignments were considered for phylogenetic analysis: (i) the sequences of the 18S rDNA gene, (ii) the sequences of the gGAPDH gene, and (iii) the concatenated sequences of the 18S rDNA and gGAPDH genes. MLSTest software (v1.0.1.23) [34] was used to concatenate alignments.

Phylogenetic analysis for both genes was performed by the maximum likelihood (ML) method, using MEGA, and the Bayesian method, using MrBayes v.3.2.7a [35], in the CIPRES platform [36]. The trees made by ML were constructed with 1000 Bootstrap repetitions. The best substitution model was selected using the AIC criterion in MEGA. The tree for the 18S rDNA gene was performed using the Kimura two parameters substitution model [37]. The Tamura-three parameters substitution model was used [38] for phylogenetic inference in the gGAPDH gene. Finally, the time-reversible general substitution model was used for maximum likelihood inference on concatenated alignments. All models were used with a discrete Gamma (+G) distribution of the substitution rate between sites and assuming the existence of a fraction of invariable sites (+I). The HKY85 model [39] with an invariant site ratio and gamma distribution was used for the concatenated alignments and for the individual genes, which were run for 1,000,000 generations, with trees sampled every 100 generations, and 25% of the early sample trees were discarded as burn-in.

In this study, and considering most studied trypanosomes, we used a phylogenetic criterium to define species boundaries. At least 1% of genetic distance to any other species was considered as the cutoff to propose a novel species. Model p-corrected distances between isolates were calculated using the MEGA program.

## Figures and Tables

**Figure 1 pathogens-09-00731-f001:**
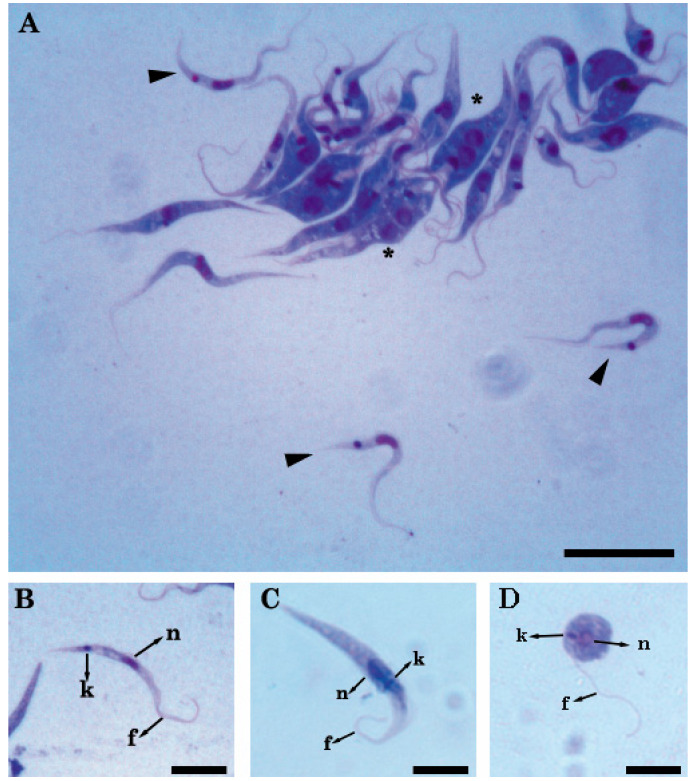
Microphotography at 1000× of parasites in cultures with May Grünwald and Giemsa. (**A**) Exponential phase culture: trypomastigotes (arrow) were observed among the epimastigotes; the dividing epimastigotes are denoted with an asterisk; (**B**) Trypomastigote; (**C**) Epimastigote; (**D**) Spheromastigote. Nucleus = n, kinetoplast = k, flagellum = f. Scale bars: (**A**) 20 µm and (**B**–**D**) 10 µm.

**Figure 2 pathogens-09-00731-f002:**
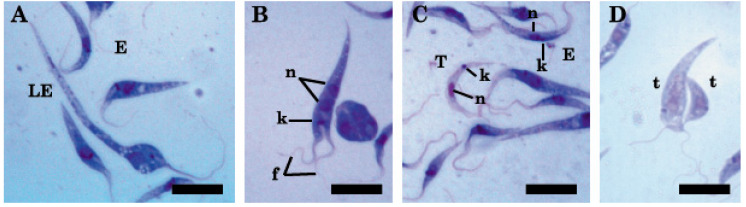
Microphotography at 1000× of parasites in cultures with May Grünwald and Giemsa. (**A**) The difference in sizes between a long epimastigote (LE) and a simple epimastigote (E) is observed; (**B**) Dividing epimastigote with two nuclei, two flagella, and one kinetoplast; (**C**) Trypomastigote (T) smaller than epimastigote (E); (**D**) Transitional form (t). Nucleus = n, kinetoplast = k, flagellum = f. Scale bar (**A**–**D**) 10 µm.

**Figure 3 pathogens-09-00731-f003:**
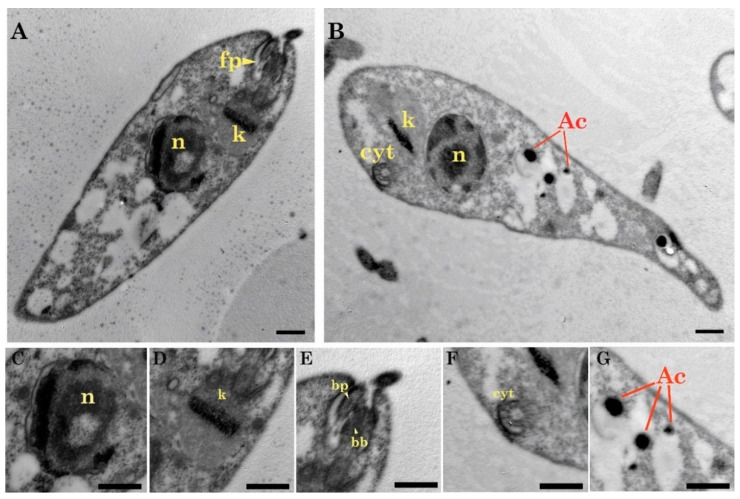
Electron micrograph of the exponential growth phase. (**A**) Epimastigote with a tapered shape, central nucleus (n), anterior kinetoplast (k) in a position close to the flagellar pocket (fp); (**B**) epimastigote showing the position of the nucleus (n), kinetoplast (k), cytostome (cyt) and acidocalcisomes (Ac); (**C**) Nucleus (n) with chromatin spots on the inner side of the nuclear membrane; (**D**) Rod shaped kinetoplast (k); (**E**) Structure of the flagellum: basal body (bb), probasal body (pb), and cross section of the flagellum; (**F**) Cross section cytostome (cyt); (**G**) Acidocalcisomes (ac). Scale bar (**A**–**B**) 1 µm; (**C**–**G**) 0.5 µm).

**Figure 4 pathogens-09-00731-f004:**
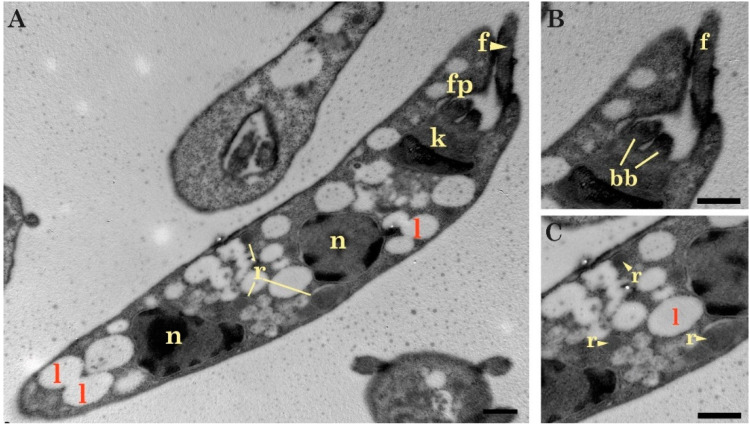
Transmission electron micrograph of the isolated parasites in the exponential phase. (**A**) Division cell with two nuclei (n), a kinetoplast (k), and a single flagellar pocket (fp); (**B**) Flagellar pocket with two basal body (bb); (**C**) Numerous vesicular bodies highlighting lipid bodies (l) and reservosomes (r). Scale bar (**A**–**C**) 0.5 µm.

**Figure 5 pathogens-09-00731-f005:**
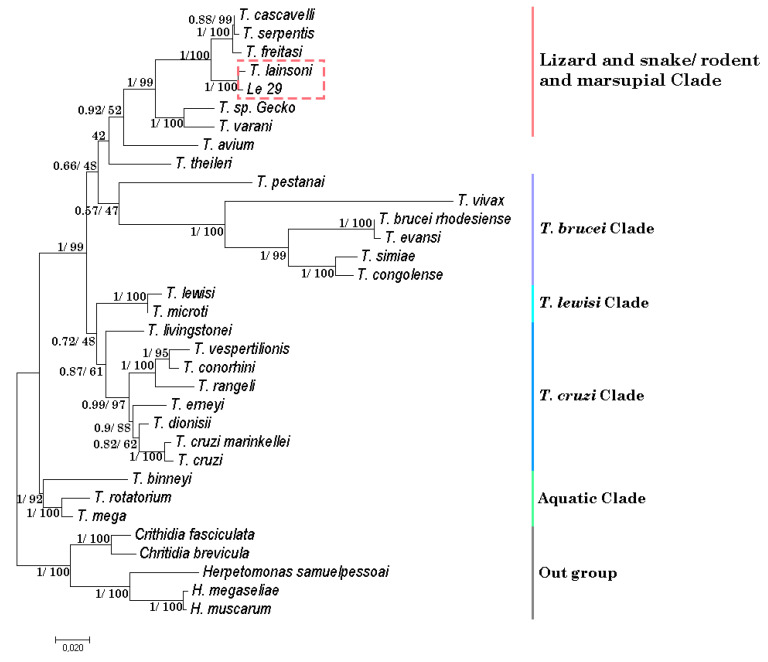
Phylogenetic analysis performed by the maximum likelihood method of the concatenated 18S rDNA and gGAPDH genes in 33 species of trypanosomatids. Species of the genera *Herpetomonas* and *Crithidia* were included as out groups. The two values at nodes represent: Bayesian analysis support/ML method bootstrap value. *Trypanosoma lainsoni* and Le29 are found within the LSRM clade.

**Table 1 pathogens-09-00731-t001:** Measurements of the different forms observed by light microscopy. Mean ± Standard Deviation.

Form	Body Average Length	Free Average Flagellum	Total Average Length	Average Width
Trypomastigote	18.15 ± 6.26	8.67 ± 3.02	28.13 ± 6.74	1.5 ± 0.47
Epimastigote	21.88 ± 3.28	11.43 ± 3.81	33.61 ± 6.08	2.3 ± 0.8
Long Epimastigote	44.25 ± 7.22	17.13 ± 6.32	61.85 ± 13.51	3.84 ± 1.55
Transitional form	13.98 ± 6.76	12.85 ± 3.52	27.26 ± 9.69	4.19 ± 1.34
Spheromastigote	6.12 ± 1.7	16.27 ± 5.19	23.01 ± 5.64	4.29 ± 1.8

**Table 2 pathogens-09-00731-t002:** This table shows the location codes of the sequences in GenBank.

**Partial Sequence**	
Le29	MT363779
Ca37	MT363780
Ca47	MT363781
**Complete Sequence**	
Le29	MT373343/MT373412
Ca37	MT363778/MT373413
Ca47	MT363777/MT373414

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
