# Peer review of "A Novel Genotype and First Record of Trypanosoma lainsoni in Argentina"

_pathogens, 2020, doi:10.3390/pathogens9090731_

Round 1

Reviewer 1 Report

The study demonstrates the occurrence of the different genotype of T. lainsoni in rodents (Calomys sp.) and wild cat (Leopardus geoffroyi) inhabiting the same (?) area of Argentina. This finding is remarkable, but, to be honest, I think it's not interesting enough for a journal like Pathogens. However, authors as an addition to this observation bring some more morphological characterization (TEM and OM for hemoculture stages) of this species. This could be a significant contribution to the current knowledge; however, the quality of TEM microphotographs is rather low and insufficient. Also, a photo of blood form (from blood smears) is missing. This part must be improved.

Minor comments:

Abstract: Calomys sp. as a host of T. lainsoni must be also mentioned in abstract

27 rADN

32 Kinetoplastea

36 “all classes of vertebrate hosts” instead of “several vertebrates”

37 invertebrates (not only arthropods)

40-1 also veterinary important trypanosomes are very well studied!

50 No trypanosomes from birds, fish, or amphibians?

Fig 1: D – spheromastigote – it is not clear if the photo represents the mentioned morphotype (it should be also just some artifact)

Fig 3-4 (TEM): rather low quality, ultrastructures are poorly visible, etc.

Table 2, 3, and 5 – Supplementary material

(BTW, Table 4 (Hosts and geographic origin of trypanosomatids and Genbank accession numbers of

267 sequences included in this study) should be Table 5).

Fig 5: Both mentioned monoxenous trypanosomatids, Leptomonas peterhoffi and Walaceina brevicula, belong to the species Crithidia brevicula !!!  

181 Table A1 ?

Reviewer 2 Report

Diaz and colleagues present a straight-forward morphometric and molecular characterization of a Trypanosoma isolate that they identify as T. lainsoni.  Overall, the manuscript is well written and there are only a few points to address. 

  1. I am not convinced of the significance of geopolitical boundaries (one isolate from Brazil; three isolates from Argentina) as being important in the significance of the isolates under study. Geological, overlapping host, and vectorial similarities are likely a better set of criteria to define the range.

  1. ‘little’ would be a better word than ‘few’

  1. Where does the number 36.4 um for ‘similar size’ come from? None of the five forms presented in Table 2 have this size.

  1. The authors raise a big issue for these types of study – what percentage DNA sequence similarity defines the same species, and what percentage DNA sequence similarity defines different species? For readers with less experience in molecular phylogenetic terms, the text could be revised to better explain the differences and similarities between interpretations based on percentage similarity/bootstrap values and genetic distance.  A bold statement of the criteria that the authors use to define same/different species should be included in the text.  The authors are probably correct in their designation, however to play the devil’s advocate it could as easily be argued (based on genetic distances) that theor isolates are a different species.

Appendix A.  What do the colored boxes mean.  Colors should be defined in the legend.

Round 2

Reviewer 1 Report

I have just a very few (minor) comments:

Figure 2: C: nucleus (n) and kinetoplast (k) should be pointed (in the epimastigote form some small dot on the apical end of the cell is visible (but (k) is near to the nucleus)

Figure 4: A: K vs k, N vs n, bb is missing (bb is on B), why l is in red, C: bb is missing 

Figure 5 (etc.) - I would suggest presenting like 0.99/97 (Bayesian probability/ML bootstrap) etc., not red and black 

Figure 5: Herpetomonas, Crithidia, Leptomonas and Wallaceina - use Italic; BUT !!! Wallaceina and Leptomonas are not presented any more in the Figure; on the other hand, Leishmania is not mentioned as an outgroup (the same for other figures !!!)

Table A3: Esmeraldo vs Esmeraldo; T. cruzi Silvio vs T. cruzi Silvio, Silvio is not in the legend; T. cruzi marinkellei is a subspecies, not a strain!

Table A5: Genbank vs GenBank

Table A5: sp vs sp.; Bos Taurus vs Bos taurus; G. vs Glossina; Tadarida vs Tadarida; Midge - is it really impossible to find the true host (probably some mosquito species...)

Fig. A2: why Trypanosoma lainsoni and Le29 are not highlighted as in other figures?

Author Response

Consulte el archivo adjunto.
